# Photoelectrochemical Response Enhancement for Metallofullerene-[12]Cycloparaphenylene Supramolecular Complexes

**DOI:** 10.3390/nano12091408

**Published:** 2022-04-20

**Authors:** Jie Zhang, Ling Qiu, Linshan Liu, Yang Liu, Peng Cui, Fang Wang, Zhuxia Zhang

**Affiliations:** 1College of Chemistry, Taiyuan University of Technology, Taiyuan 030024, China; zhangjie654@iccas.ac.cn (J.Z.); ql7222785@163.com (L.Q.); l_lynne33@163.com (L.L.); liuliuyang258@163.com (Y.L.); 2Beijing National Laboratory for Molecular Sciences, Key Laboratory of Molecular Nanostructure and Nanotechnology, Institute of Chemistry, Chinese Academy of Sciences, Beijing100190, China; 3University of Chinese Academy of Sciences, Beijing 100049, China; 4School of Information, Guizhou University of Finance and Economics, Guiyang 550025, China; 5Collaborative Innovation Center for Shanxi Advanced Permanent Magnetic Materials and Technology, Linfen 041004, China; wangfanghc@sxnu.edu.cn

**Keywords:** metallofullerene, cycloparaphenylene, supramolecular, photoelectrochemical, photocurrent

## Abstract

The photoelectrochemical properties of three metallofullerene-[12]cycloparaphenylene ([12]CPP) supramolecular complexes of Sc_3_N@C_78_⊂[12]CPP, Sc_3_N@C_80_⊂[12]CPP, and Sc_2_C_2_@C_82_⊂[12]CPP were studied. It was revealed that the photocurrent responses of these supramolecular complexes show enhancement compared with those of pristine metallofullerenes, indicating the efficient photocurrent generation and promoted charge carrier transport caused by the supramolecular interaction. The results show that Sc_2_C_2_@C_82_ and Sc_2_C_2_@C_82_⊂[12]CPP have the strongest photocurrents. Then, by comparing the photocurrent intensities of the metallofullerene-biphenyl derivates mixture and the metallofullerene⊂[12]CPP complexes, it was demonstrated that the host–guest interaction is the key factor promoting photocurrent enhancement. At the same time, by observing the microscopic morphologies of pristine fullerenes and supramolecular complexes, it was found that the construction of supramolecules helps to improve the morphology of metallofullerenes on FTO glass. Additionally, their electrical conductivity in optoelectronic devices was tested, respectively, indicating that the construction of supramolecules facilitates the transport of charge carriers. This work discloses the potential application of metallofullerene supramolecular complexes as photodetector and photoelectronic materials.

## 1. Introduction

Endohedral metallofullerenes have attracted great interest from scientists due to their special applications in semiconductors and electronic devices with respect to their combining characters from fullerene cage and internal metal cluster [1,2]. Because of the electron transfer from endohedral metal atoms to outer fullerene cage, the electronic properties of metallofullerenes are changed drastically from those of empty fullerenes [3,4]. Metallofullerenes have varied HOMO–LUMO gaps and unique photochemical properties [5,6]; therefore, metallofullerenes are potential nanomaterials with wide application in field-effect transistors [7], photodetectors [8], photovoltaic devices [9], etc. For example, Kobayashi et al. reported bimetal endohedral fullerene La_2_@C_80_ thin film as n-type field effect transistor [10]. Moreover, in 2017, Lu’s group discovered that Lu_2_@C_82_ nanorods have enhanced photoluminescence and optoelectronic properties [11]. It should be noted that the electronic properties of metallofullerenes need to be improved due to the spherical form, which would influence carrier mobility.

Host–guest supramolecular construction is an effective method for modulating the state and properties of metallofullerenes, as there are strong π–π interaction between them [12]. For host molecules, functional macrocyclic molecules have always played a very important role in the field of supramolecular chemistry [13]. Compared with other host molecules, the cycloparaphenylene (CPP) carbon nanorings are very suitable for metallofullerenes, as their supramolecular structures have high stability [14]. For example, our group has studied the paramagnetic properties of C_80_-based metallofullerenes within [12]CPP, revealing the confinement effect on the metallofullerene spin modulation [15]. In 2019, Delius et al. successfully carried out the synthesis of a porphyrin–[10]CPP conjugate and studied its strong association with a range of fullerenes and demonstrated that [10]CPP as a supramolecular junction enables efficient charge transport between a porphyrin electron donor and unmodified fullerene electron acceptors [16]. These results, together with Yamago’s recent report on the thin-film conductivity of [10]CPP and its alkoxy derivatives [17], imply that supramolecular complexes of [10]CPP and fullerenes may be a useful addition to the toolbox of organic electronics. In addition, Yang et al. studied present syntheses and characterizations of two novels CPP-like curved nanographene that strongly bind with fullerene C_60_ to form photoconductive heterojunctions [18]. The results indicate that there is a fast photoinduced electron transfer process in the supramolecular heterojunction. Recently, Zhao et al. constructed a double-walled carbon nanoring supramolecular [6]CPP-[12]CPP by self-assembly and found that [6]CPP⊂[12]CPP shows highly enhanced photoconductivity and photocurrent under light irradiation compared with those of pristine monomers [19]. However, the optoelectronic property of metallofullerenes within the [12]CPP supramolecular complex has not been studied. Considering that the type of supramolecular system may change the molecular energy levels [20], control the metallofullerene assembly [17], and consequently influence the carrier mobility [8,18], therefore, it is significant to explore the optoelectronic property of this new kind of supramolecular complex.

In this paper, Sc_3_N@C_78_, Sc_3_N@C_80_, and Sc_2_C_2_@C_82_ were selected to study the photoelectrochemical property of their metallofullerene-[12]CPP supramolecular complexes. The metallofullerene-[12]CPP film was prepared by spin coating on fluorine-doped tin dioxide (FTO) glass as the working electrode, and the photoconductive measurement was performed on an electrochemical system. The photocurrent responses of these supramolecular complexes were compared.

## 2. Materials and Methods

Metallofullerene Sc_3_N@C_78_, Sc_3_N@C_80_ were synthesized by arc-discharge method. First, the graphite powder and graphite rod were vacuum-dried at 100 °C for 12 h, and then the graphite powder and Sc/Ni alloy were mixed uniformly in a mass ratio of 1:3 and filled into hollow graphite rods. The graphite rods were then placed in a VD-250 vacuum arc furnace and energized by arc discharge under a pressure of 180 Torr He and 20 Torr N_2_. The ashes obtained after cooling were Soxhlet extracted with toluene solvent for 24 h to obtain a toluene solution of fullerenes. The synthesis method of Sc_2_C_2_@C_82_ is similar to that of Sc_3_N@C_78_ and Sc_3_N@C_80_; the difference is that 200 torr He needs to be added when adding gas, and N2 is not required. Then, pure metallofullerenes were isolated and purified by high performance liquid chromatography (HPLC). The [12]CPP, biphenyl, p−terphenyl, and p−quaterphenyl was purchased from J&K.

Metallofullerene solution with a concentration of 4 × 10^−5^ M was prepared in *o*-DCB solution. The complexes of Sc_3_N@C_78_, Sc_3_N@C_80_, and Sc_2_C_2_@C_82_ with [12]CPP were mixed with a 1:1 mole ratio. For the film preparation of metallofullerene complex, 10 μL sample was dropped to the FTO glass, and the spin-coating was carried out with a spin-speed of 1800 rpm for 50 s.

UV–vis absorption spectra were acquired from a HITACHI UV/VIS/NIR Spectrometer UH4150.

Fluorescence quenching experiments were recorded on a fluorescence spectrometer (F-7100, HITACHI) with excitation at 350 nm in toluene solution. Method: To toluene solution of (4.0 × 10^−7^ M, 2 mL) [12]CPP was added 20 μL of metallofullerene samples (4.0 × 10^−5^ M).

AFM experiments were performed using a Multimode VIII atomic force microscopy (Bruker Inc., Billerica, MA, USA). SEM is a module that comes with the AFM device. The measurements are under room temperature and atmospheric pressure.

We prepared 100 μL (4 × 10^−3^ M) samples of Sc_2_C_2_@C_82_ and Sc_2_C_2_@C_82_⊂[12]CPP, respectively. FTO substrates were cleaned ultrasonically several times in sequence baths of acetone, ethyl alcohol, and deionized water for 20 min. The solution was quickly spin-coated on an FTO glass electrode (100 μL, 1800 rpm min^−1^) and aged for 50 s at the temperature of 80 °C. The model of the spin coater is WS-650MZ-23NPPB, manufactured by Laurell Technologies. The photocurrent measurement was carried out using the CHI650E electrochemical workstation. The device of FTO Sc_2_C_2_@C_82_ and Sc_2_C_2_@C82⊂[12]CPP was conducted to measure the I–V curve by the ORIEL IQE-200 system with an AM 1.5 solar spectrum filter and a Keithley 2420 source meter.

Geometry optimization and excitation states were all calculated at the level of B3LYP/6-31G* and realized by DFT. The Gaussian 16 (A.03) program package was employed for all quantum chemistry calculations.

## 3. Results and Discussion

Firstly, we synthesized Sc_3_N@C_78_-*D_5h_*, Sc_3_N@C_80_-*I_h_*, and Sc_2_C_2_@C_82_-*C_s_* synthesized by the arc-discharge method and isolated by HPLC. Metallofullerene and [12]CPP mildly were mixed with 1:1 mole ratio at room temperature to obtain the supramolecular complexes of Sc_3_N@C_78_⊂[12]CPP, Sc_3_N@C_80_⊂[12]CPP, and Sc_2_C_2_@C_82_⊂[12]CPP. Figure 1 shows the theoretically optimized structure of these complexes. MALDI-TOF mass spectrometry demonstrated stable formation of the three complexes [21,22,23]. In Appendix A, the mass peak of 1998 represents the molecular ion peak of Sc_3_N@C_78_⊂[12]CPP, and the mass peak of 1085 corresponds to the fragments of Sc_3_N@C_78_. In Appendix A, the mass peak of 2022 represents the molecular ion peak of Sc_3_N@C_80_⊂[12]CPP, and the mass peak of 1109 corresponds to the fragments of Sc_3_N@C_80_. In addition, in Appendix A, the mass peak of 2011 represents the molecular ion peak of Sc_2_C_2_@C82⊂[12]CPP, and the mass peak of 1098 corresponds to the fragments of Sc_2_C_2_@C_82_. Subsequently, we performed UV-near infrared visible (UV-Vis) characterization of Sc_3_N@C_78_, Sc_3_N@C_80_, and Sc_2_C_2_@C_82_ and their respective complexes with [12]CPP (Appendix A).

The film was firstly prepared by spin coating on FTO glass as the working electrode (Figure 1), then the photoelectrochemical experiments of the three pristine metallofullerenes were carried out.

It can be seen from Figure 2a that the metallofullerenes with the same molar concentration were irradiated by natural light for 20 s every 20 s by the 1 V bias voltage, and it was found that the corresponding responses were generated. It can be clearly seen that immediately after irradiation, a stable anode photocurrent was generated and decreased without irradiation. These processes can be repeated reversibly multiple times, which indicate that surface active film is sufficiently stable during light exposure. The light response intensity of Sc_2_C_2_@C_82_ was the highest at about 0.2 μA (Figure 2b), and those of Sc_3_N@C_78_ and Sc_3_N@C_80_ were around 0.1 μA, see Figure 2a, revealing the metallofullerene-dependent photoconductive property [3,24]. These results show the high light absorption ability of Sc_2_C_2_@C_82_, as revealed in its absorption of the UV-Vis-NIR spectrum (Appendix A). It can be seen that Sc_2_C_2_@C_82_ still has an absorption peak at a wavelength of 1200 nm, indicating that Sc_2_C_2_@C_82_ can produce energy-level transitions under low-energy illumination.

Then, the photoelectrochemical experiments of three metallofullerenes-[12]CPP complexes were studied, as shown in Figure 2b. The results show that the photocurrent responses of the metallofullerenes-[12]CPP complexes show an enhancement as compared with pristine metallofullerenes, indicating the efficient photocurrent generation and promoted charge carrier transport processes caused by the host–guest interaction. Notably, the Sc_2_C_2_@C_82_⊂[12]CPP still had the strongest light response intensity (0.32 μA) compared to those of Sc_3_N@C_78_⊂[12]CPP and Sc_3_N@C_80_⊂[12]CPP. Different fullerenes and CPPs have different host–guest interactions, which may be one of the reasons for this difference [20].

The HOMO–LUMO energy gaps of metallofullerene Sc_3_N@C_78_ (2.269 eV), Sc_3_N@C_80_ (2.529 eV), Sc_2_C_2_@C_82_ (1.394 eV) (Figure 3a) and complex Sc_3_N@C78⊂[12]CPP (2.273 eV), Sc_3_N@C_80_⊂[12]CPP (2.538 eV), and Sc_2_C_2_@C82⊂[12]CPP (1.406 eV) (Figure 3b) were calculated from the edges of their absorption bands (Figure 3). The HOMO–LUMO energy gap is often used to predict the electrical conductivity of a molecule [25]. It is generally considered that the smaller the HOMO–LUMO gap, the better the electrical conductivity. It can be seen that the pristine Sc_2_C_2_@C_82_ and complex Sc_2_C_2_@C82⊂[12]CPP have the smallest gap, respectively, indicating that they have the best electrical conductivity and thus the strongest photocurrent. Additionally, we note that the difference in the gap between fullerenes and complexes is not significant. The molecular orbitals were little affected for metallofullerene and CPP upon complexation. The calculation results indicate that no significant orbital interactions took place between fullerene and [12]CPP.

It is worth noting that our previous study found that the host–guest interactions between different carbon cages and CPP were different [19]. It was found by calculation that the binding energy of Sc_2_C_2_@C_82_⊂[12]CPP was the strongest at about –48.37 Kcal·mol^−1^ (Appendix A). Therefore, we have reason to believe that the difference in this interaction is one of the factors affecting the photocurrent signal.

[12]CPP has excellent fluorescence properties, and fullerenes with different sizes have different quenching abilities than [12]CPP [23]. We used these three metallofullerenes with the same concentration to quench the fluorescence of [12]CPP and obtained different fluorescence quenching efficiencies; see Figure 4. After comparison, it can be seen that Sc_2_C_2_@C_82_ has the best quenching effect on [12]CPP—about 57.7%. Combined with the conclusions given in the literature [23], it shows that Sc_2_C_2_@C_82_ has stronger host–guest interaction with [12]CPP and that this π-conjugated host–guest interaction enhanced action helps to facilitate charge carrier transfer, resulting in improved photocurrent signal [18].

In addition, we compared the photoconductive properties of Sc_3_N@C_78_, Sc_3_N@C_80_, and Sc_2_C_2_@C_82_ complexed with biphenyl, p−terphenyl, p−quaterphenyl, and [12]CPP to further disclose the effect of the host–guest interaction on the photocurrent response, as shown in Figure 5. The metallofullerenes maintain the same molar concentration for all of the photoelectrochemical experiments. The small aromatic molecules were mixed with three metallofullerene solutions at certain concentration to ensure that they had the same number of benzene rings with [12]CPP. Firstly, the photoconductive properties of the pristine biphenyl, p−terphenyl, and p−quaterphenyl were measured (Appendix A). The photocurrent responses of the three small aromatic molecules were similar. Subsequently, after comparing the metallofullerenes complexed with three small aromatic molecules and [12]CPP, it was found that the small aromatic molecules did not increase the photocurrent intensity of metallofullerenes, but the photocurrent responses of the supramolecular complexes with [12]CPP became stronger. At the same time, we characterized the UV-Vis of these biphenyls molecules and metallofullerene mixtures, and it can also be seen that the simple mixture of biphenyls and metallofullerene had not conjugated each other, as shown in Appendix A. These results further reveal that biphenyl derivatives have little effect on the photocurrent of fullerenes and that the host–guest interaction after the recombination of fullerenes and [12]CPP is the main factor for the improvement in the photocurrent response of fullerenes.

In the above experiments, the light was switched on and off six times for about 240 s, and the photocurrent was tested. It can be seen that the photocurrent signal was stable, so the FTO film had sufficient stability [19]. In detail, the surface of the metallofullerene complex films were characterized by SEM and AFM. Figure 6 shows that the film depth values of Sc_3_N@C_78_⊂[12]CPP, Sc_3_N@C_80_⊂[12]CPP and Sc_2_C_2_@C_82_⊂[12]CPP were about −79 to 62.3 nm, −80.1 to 62.3 nm, and −81.3 to 62.8 nm, respectively. The three supramolecular complex films exhibited uniform and flat surfaces. It should be noted that the surfaces of the pristine metallofullerenes have obvious aggregation (Appendix A). Therefore, the host of [12]CPP can improve the metallofullerene assembly on the FTO glass surface, and then enhance the carrier mobility of supramolecular complexes. In recent years, Wan’s team has achieved microscale superlubricity of fullerene derivatives by constructing regular host–guest assembly structures, proving that the construction of host–guests can help reduce the friction coefficient of fullerenes and improve the nanomolecular planar morphology [26]. Similar results were also reported in the STM images of Y_3_N@C_80_⊂[12]CPP on Au(111) surface, in which the Y_3_N@C_80_⊂[12]CPP randomly absorb on the surface without aggregation, as the [12]CPP nanoring can change the weak van der Waals forces and influence the self-assembly character of Y_3_N@C_80_ [15].

The supramolecular formation will change the metallofullerene assembly on the FTO glass surface, and then enhance the carrier mobility of supramolecular complexes. The I–V measurements for Sc_2_C_2_@C_80_, Sc_2_C_2_@C_80_@[12]CPP, and Sc_2_C_2_@C_80_-biphenyl derivatives mixture were then conducted. A device with a structure of FTO/samples/Ag was designed to measure the I–V curves with a standard xenon-lamp-based solar simulator [19]. As shown in Figure 7, a strong linear relationship between the current and the applied voltage ranging from −1 V to 1 V was recorded, indicating an ohmic behavior of the electrical conduction. By comparing the line slopes, it can be seen that the Sc_2_C_2_@C_82_⊂[12]CPP has a greater conductivity than the pristine Sc_2_C_2_@C_82_ and Sc_2_C_2_@C_82_-biphenyl derivatives mixtures. The shape and functional group complementarity between two electronic components form a robust host–guest complex that can be readily processed to generate a superior electronic material. This indicates that electronic communication between aromatic surfaces is particularly strong due to their tight supramolecular assembly [27]. There are many reports in the literature on the enhancement of the photocurrent performance of materials by improving the structure of thin films. Suppressing the impurity scattering associated with the structural disorder therefore increased the carrier mobility, which led to the significant increase in photocurrent observed in the photodetector [28,29,30]. Therefore, it is plausible that this improved the planar self-assembly behavior of the material by constructing supramolecules to influence the photocurrent of the material. This suggests their potential application in the field of photodetectors and optoelectronic devices.

## 4. Conclusions

In conclusion, the photoelectrochemical property of three metallofullerene⊂[12]CPP supramolecular complexes of Sc_3_N@C_78_⊂[12]CPP, Sc_3_N@C_80_⊂[12]CPP, and Sc_2_C_2_@C_82_⊂[12]CPP were studied. The results show that the photocurrent responses of the metallofullerenes-[12]CPP complexes show an enhancement as compared with pristine metallofullerenes due to the supramolecular interaction. In addition, the metallofullerene and benzene series mixtures were also comparatively investigated to illustrate the promoting effect of host–guest interaction on the photocurrent generation in metallofullerene. Through a micro-morphological characterization, it can be seen that the formation of supramolecules on the FTO glass improves the assembly morphology of the fullerene and that the I–V curve test proves that the composite has stronger electrical conductivity. These results indicate the efficient photocurrent generation and promoted charge carrier transport processes by the supramolecular structure. Consequently, this host–guest effect promotes the photocurrent responses of the metallofullerenes supramolecular complexes, illustrating their potential application in photodetector and photoelectronic devices.

## Data Availability

The data that support the findings of this study are available from the corresponding author upon reasonable request.

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
