# Peer review of "Photoelectrochemical Response Enhancement for Metallofullerene-[12]Cycloparaphenylene Supramolecular Complexes"

_nanomaterials, 2022, doi:10.3390/nano12091408_

Round 1

Reviewer 1 Report

Please find the attached document.

Author Response

This is a well written, interesting, and useful contribution I think.

My comments are as follows.

C1; The description of "2. Materials and Methods" is insufficient. You should clarify what you got from others and what you created yourself. Especially, the arc-discharge method is difficult to understand.

Response: Thanks for the valuable comment. We describe the experimental materials and methods in further detail. Sc3N@C78 and Sc3N@C80 were synthesized by the arc discharge method. First, the graphite powder and graphite rod were vacuum-dried at 100°C for 12 h, and then the graphite powder and Sc/Ni alloy were mixed uniformly in a mass ratio of 1:3 and filled into the hollow graphite rod. The graphite rods were then placed in a VD-250 vacuum arc furnace and energized by arc discharge under a pressure of 180 Torr He and 20 Torr N2. The ashes obtained after cooling were soxhlet extracted with toluene solvent for 24 hours to obtain a toluene solution of fullerenes. The synthesis method of Sc2C2@C82 is similar to that of Sc3N@C78 and Sc3N@C80, the difference is that 200 torr He needs to be added when adding gas, and N2 is not required.

Fluorescence quenching experiments were recorded on a fluorescence spectrometer (F-7100, HITACHI) with excitation at 350 nm in toluene solution. Method: To toluene solution of (4.0 × 10−7 M, 2mL) [12]CPP was added 20 μL of metallofullerene samples (4.0 × 10−5 M).

C2; From Figures S1 and S2, please explain in a little more detail why the structure could be determined to be the theoretically optimized structure in Figure 1 using MALDI-TOF and UV-Vis.

Response: Thanks for the valuable comment. In fact, mass spectrometry is a commonly used method to characterize supramolecules (Chem. Commun. 2019, 77, 11511-11514.; Carbon 2020, 161, 694-701). In Figure S1a, the mass peak of 1998 represents the molecular ion peak of Sc3N@C78⊂[12]CPP, and the mass peaks of 1085 correspond to the fragments of Sc3N@C78. In Figure S1b, the mass peak of 2022 represents the molecular ion peak of Sc3N@C80⊂[12]CPP, and the mass peaks of 1109 correspond to the fragments of Sc3N@C80. And in Figure S1c, the mass peak of 2011 represents the molecular ion peak of Sc2C2@C82⊂[12]CPP, and the mass peaks of 1098 correspond to the fragments of Sc2C2@C82. However, UV-Vis spectra of monomers and complexes are only provided in SI as a characterization tool.

C3; Please show sufficient cause or references for the conclusions stated on page 3, lines 113-115.

Response: Thanks for the valuable comment. The light response intensity of Sc2C2@C82 is the highest about 0.2 μA (Figure 2b), and those of Sc3N@C78 and Sc3N@C80 are around 0.1 μA, see Figure 2a, revealing the metallofullerene-dependent photoconductive property (J. Phys. Chem. B 2001, 39, 9406-9412.; Jpn J Appl Phys. 2002, 41, 2254-2255).

C4; Please show sufficient cause or references for the conclusions stated on page 6, lines 181-186.

Response: Thanks for the valuable comment. In recent years, Wan's team has achieved microscale superlubricity of fullerene derivatives by constructing regular host-guest assembly structures, proving that the construction of host-guests can help reduce the friction coefficient of fullerenes and improve the nanomolecular planar morphology (ACS Appl. Mater. 2020, 16, 18924-18933).

C5; Please explain Figure 6 in more detail.

Response: Thanks for the valuable comment. The I–V measurements for Sc2C2@C80, Sc2C2@C80@[12]CPP, and Sc2C2@C80-biphenyl derivatives mixture were then conducted. A device with a structure of FTO/samples/Ag was designed to measure the I–V curves with a standard xenon-lamp-based solar simulator (Nanoscale 2021, 9, 4880-4886). As shown in Figure 7, a strong linear relationship between the current and the applied voltage ranging from −1 V to 1 V was recorded, indicating an ohmic behavior of the electrical conduction. By comparing the line slopes, it can be seen that the Sc2C2@C82⊂[12]CPP has a greater conductivity than the pristine Sc2C2@C82 and Sc2C2@C82-biphenyl derivatives mixtures.

C6; The discussion on page 6, lines 203-206 seems to be a leap forward. Please describe the relationship between the thin film structure and absorption / emission in a little more detail

Response: Thanks for the valuable comment. There are many reports in the literature on the enhancement of the photocurrent performance of materials by improving the structure of thin films. Suppressing the impurity scattering associated with the structural disorder and therefore increased the carrier mobility, which led to the significant increase in photocurrent observed in the photodetector (Nanoscale Res. Lett. 2020, 1, 47.; IEEE Photon. J. 2019, 6, 1-8.; Vacuum 2017, 140, 106-110.). Therefore, it is plausible that this improvement of the planar self-assembly behavior of the material by constructing supramolecules to influence the photocurrent of the material. This suggests their potential application in the field of photodetectors and optoelectronic devices.

Reviewer 2 Report

This study has been focused on the PEC response of metallofullerene-cycloparaphenylene complxes with different fullerenes and ascribe the enhancement of PEC signals to host-guest interaction as the key factor. But authors should give detailed descriptiuon on the kinds of interaction. They also should give detailed explanation on the results to support the listed results with data. HOMO-LUMO electron density distribution and the HOMO-LUMO transition reflects the interaction between Sc metal and fullerene. It is very difficult to understand clearly on the relationship between smaller HOMO-LUMO energy gap and better electrical conductivity simply. Authors shouls give answer and describe in the the text on the following points

  1. Give PL data
  2. 2. Photoconductivity data of metallofullerene complexs with biphenyl, p-terphenyl and p-quaterphenyl are measured in solution state or film state
  3. 3. Give caption for Scheme 1
  4. 4. In the introduction, authors should explain more specifically on the reason that metallofullerene electricity and CPP were chosen
  5.  They also give data and explanation on the stability of the film
  6. Some grammer mistakes : page 2 line 80; page 2 line 93; page 3 line 99 and 109. 

Author Response

Responses to the reviewer’s comments

Referee: 2
This study has been focused on the PEC response of metallofullerene-cycloparaphenylene complxes with different fullerenes and ascribe the enhancement of PEC signals to host-guest interaction as the key factor. But authors should give detailed descriptiuon on the kinds of interaction. They also should give detailed explanation on the results to support the listed results with data. HOMO-LUMO electron density distribution and the HOMO-LUMO transition reflects the interaction between Sc metal and fullerene. It is very difficult to understand clearly on the relationship between smaller HOMO-LUMO energy gap and better electrical conductivity simply. Authors shouls give answer and describe in the the text on the following points

Response: Thanks for the valuable comment. The effect of HUMO-LUMO gap value on photoconductivity mentioned in the manuscript is from the literature: RSC Adv., 2013,3, 25881-25890. And URL research: http://sobereva.com/543

Commont 1: Give PL data

Response: Thanks for the valuable comment. [12]CPP has excellent fluorescence properties, and fullerenes with different sizes have different quenching abilities to [12]CPP. We used these three metallofullerenes with the same concentration to quench the fluorescence of [12]CPP, and obtained different fluorescence quenching efficiencies, see Figure 4. After comparison, it can be seen that Sc2C2@C82 has the best quenching effect on [12]CPP, about 57.7%. Combined with the conclusions given in the literature (Carbon 2020, 161, 694-701), it shows that Sc2C2@C82 has stronger host-guest interaction with [12]CPP, and this π-conjugated host-guest interaction Enhanced action helps to facilitate charge carrier transfer, resulting in improved photocurrent signal (Angew. Chem. Int. Ed. 2019, 58, 6244–6249).

Commont 2: Photoconductivity data of metallofullerene complexs with biphenyl, p-terphenyl and p-quaterphenyl are measured in solution state or film state.

Response: Thanks for the valuable comment. We measured photoconductivity data for metallofullerene complexes with biphenyl, p-terphenyl, and p-tetraphenyl in the thin film state. The I–V measurements for Sc2C2@C80, Sc2C2@C80@[12]CPP, and Sc2C2@C80-biphenyl derivatives mixture were then conducted. A device with a structure of FTO/samples/Ag was designed to measure the I–V curves with a standard xenon-lamp-based solar simulator (Nanoscale 2021, 9, 4880-4886). As shown in Figure 7, a strong linear relationship between the current and the applied voltage ranging from −1 V to 1 V was recorded, indicating an ohmic behavior of the electrical conduction. It can be seen that the Sc2C2@C82⊂[12]CPP has a greater conductivity than the pristine Sc2C2@C82 and Sc2C2@C82-biphenyl derivatives mixtures.

Commont 3: Give caption for Scheme 1

Response: Thanks for the valuable comment. We have provided a caption for Scheme 1. This part has been given in the manuscript: Schematic diagram of supramolecular thin film preparation and photocurrent measurement.

Commont 4: In the introduction, authors should explain more specifically on the reason that metallofullerene electricity and CPP were chosen.

Response: Thanks for the valuable comment. We explain the reasons for choosing metallofullerene electros and CPPs more specifically in the manuscript. In 2019, Delius et al. successfully carried out the synthesis of a porphyrin–[10]CPP conjugate and studied its strong association with a range of fullerenes, and demonstrated that [10]CPP as a supramolecular junction enables efficient charge transport between a porphyrin electron donor and unmodified fullerene electron acceptors (Angew. Chem. Int. Ed. 2018, 57, 11549 –11553). These results, together with Yamago’s recent report on the thin-film conductivity of [10]CPP and its alkoxy derivatives (J. Am. Chem. Soc. 2017, 51, 18480-18483), imply that supramolecular complexes of [10]CPP and fullerenes may be a useful addition to the toolbox of organic electronics. In addition, Yang et al. studied presents synthesis and characterizations of two novel CPP-like curved nanographene that strongly bind with fullerene C60 to form photoconductive heterojunctions(Angew. Chem. Int. Ed. 2019, 19, 6244-6249). The results indicate that there is a fast photoinduced electron transfer process in the supramolecular heterojunction. Recently, Zhao et al. constructed a double-walled carbon nanoring supramolecular [6]CPP-[12]CPP by self-assembly and found that [6]CPP⊂[12]CPP shows highly enhanced photoconductivity and photocurrent under light irradiation compared to those of pristine monomers (Nanoscale 2021, 9, 4880-4886).

Commont 5: They also give data and explanation on the stability of the film

Response: Thanks for the valuable comment. In the above experiments, the light was switched on and off 6 times for about 240 s and the photocurrent was tested. It can be seen that the photocurrent signal is stable, so the FTO film has sufficient stability (Nanoscale 2021, 9, 4880-4886). In detail, the surface of the metallofullerene complex films was characterized by SEM and AFM. Figure 6 shows that the film depth values of Sc3N@C78⊂[12]CPP, Sc3N@C80⊂[12]CPP and Sc2C2@C82⊂[12]CPP are about -79­-62.3 nm, -80.1-62.3 nm, and -81.3-62.8 nm, respectively. The three supramolecular complex films exhibit uniform and flat surfaces.

Commont 6: Some grammer mistakes : page 2 line 80; page 2 line 93; page 3 line 99 and 109. 

Response: Thanks for the valuable comment. Grammer mistakes in the manuscript have been corrected in their corresponding places.

page 2 line 80: Prepared 100 μL (4×10−3 M) samples of Sc2C2@C82 and Sc2C2@C82⊂[12]CPP, respectively.

Revise: We prepared 100 μL (4×10−3 M) samples of Sc2C2@C82 and Sc2C2@C82⊂[12]CPP, respectively.

page 2 line 93: Sc3N@C78-D5h, Sc3N@C80-Ih, and Sc2C2@C82-Cs were firstly synthesized by arc-discharge method and isolated by HPLC.

Revise: Firstly, we synthesized Sc3N@C78-D5h, Sc3N@C80-Ih, and Sc2C2@C82-Cs by arc-discharge method and isolated by HPLC.

page 3 line 99: Subsequently, we performed UV-near infrared visible (UV-Vis) characterization of Sc3N@C78, Sc3N@C80 and Sc2C2@C82 and their respective complexes with [12]CPP (Figure S2).

Revise: Subsequently, we performed UV-near infrared visible (UV-Vis) characterization of Sc3N@C78, Sc3N@C80, and Sc2C2@C82 and their respective complexes with [12]CPP (Figure S2).

page 3 line 109: It can be seen from Figure 2a that the metallofullerenes with the same molar concentration is irradiated by natural light for 20 s every 20 s by the 1 V bias voltage, and it is found that the corresponding responses are generated

Revise: It can be seen from Figure 2a that the metallofullerenes with the same molar concentration are irradiated by natural light for 20 s every 20 s by the 1 V bias voltage, and it is found that the corresponding responses are generated.

Round 2

Reviewer 2 Report

Authors revised well according to the refree's comment. So it can ve accepted as it is.